# Environmental conditions shape the nature of a minimal bacterial genome

Magdalena Antczak [1], Martin Michaelis [1] & Mark N. Wass [1]

Of the 473 genes in the genome of the bacterium with the smallest genome generated to date, 149 genes have unknown function, emphasising a universal problem; less than 1% of proteins have experimentally determined annotations. Here, we combine the results from state-of-the-art in silico methods for functional annotation and assign functions to 66 of the 149 proteins. Proteins that are still not annotated lack orthologues, lack protein domains, and/ or are membrane proteins. Twenty-four likely transporter proteins are identified indicating the importance of nutrient uptake into and waste disposal out of the minimal bacterial cell in a nutrient-rich environment after removal of metabolic enzymes. Hence, the environment shapes the nature of a minimal genome. Our findings also show that the combination of multiple different state-of-the-art in silico methods for annotating proteins is able to predict functions, even for difficult to characterise proteins and identify crucial gaps for further development.

---

[1] Industrial Biotechnology Centre, School of Biosciences, University of Kent, Canterbury, Kent CT2 7NJ, UK. Correspondence and requests for materials should be addressed to M.M. (email: m.michaelis@kent.ac.uk) or to M.N.W. (email: m.n.wass@kent.ac.uk)

A long-term goal of synthetic biology has been the identification of the minimal genome, i.e., the smallest set of genes required to support a living organism. The bacterium with the smallest genome generated to date is based on *Mycoplasma mycoides*[1]. Its minimal bacterial genome consists of 473 genes including essential genes and a set of genes associated with growth, termed 'quasi-essential'[1]. The minimal genome study assigned function to proteins encoded by the minimal genome by considering matches to existing protein families in the TIGRFAM[2] database, genome context and structural modelling[1]. Proteins were annotated with molecular functions and grouped into 30 biological process categories (including an unclear category, where the biological process was not known). The proteins were further assigned to five classes according to the specificity and confidence of the molecular function annotations that they had been assigned: Equivalog (confident hits to TIGRFAM families), Probable (low confidence match to TIGRFAM families supported by genome context or threading), Putative (multiple sources of evidence but lower confidence), Generic (general functional information identifiable, e.g., DNA binding or membrane protein, but specific function unknown) and Unknown (unable to infer even a general function). The final two confidence classes, Unknown (65 genes) and Generic (84 genes) form the group of genes whose function is unknown. Hence, almost a third (149) of the encoded 473 proteins are of unknown function, which emphasises our limited understanding of biological systems[1].

This lack of functional annotation is not restricted to the minimal bacterial genome. One-third of protein-coding genes from bacterial genomes lack functional annotations[3]. Recent experimental approaches have begun to identify the function of 'hypothetical' proteins of unknown function[4]. However, the continual improvement of high-throughput sequencing methods has resulted in a rapid increase in the number of organisms for which genome sequences are available and the functional annotation of the encoded gene products lags behind[4]. Less than 1% of the 148 million protein sequences in UniProt[5] are annotated with experimentally confirmed functions in the Gene Ontology (GO)[6] (April 2019). To address this gap, computational methods for protein function prediction have been developed and significantly advanced over the past 15 years as demonstrated by the recent Critical Assessment of Functional Annotation (CAFA) challenges[7,8].

Here, we perform an extensive in silico analysis of the proteins of unknown function encoded by the minimal bacterial genome using an approach that combines 22 different computational methods ranging from identification of basic properties (e.g., protein domains, disorder and transmembrane helices) to state-of-the-art protein structural modelling and methods that infer GO-based protein functions, including those that have performed well in CAFA experiments.

## Results

### Orthologues for the proteins in the minimal genome. 
Hutchison et al.[1] used BLAST to identify homologues of the minimal genome proteins in a set of 14 species ranging from non-mycoides mycoplasmas to archaea. They found that while many of the proteins from the Equivalog, Probable, Putative and Generic classes have homologues in all 14 species, very few of the sequences in the Unknown class had homologues outside of *M. mycoides*, with none in *M. tuberculosis, A. thaliana, S. cerevisiae* and *M. jannaschii*.

Here, eggNOG-Mapper[9] (see methods) was used to identify orthologues for the minimal genome proteins across the three kingdoms of life. Overall the analysis showed that very few of the

Unknown class of proteins (7%) have related sequences in eukaryotes or archaea (6%) while just over half (55%) have orthologues in other bacterial species, primarily in terrabacteria, the clade that *M. mycoides* belongs to (Fig. 1a, Supplementary Fig. 1 and Supplementary Data File 1). In contrast, many of the proteins in the other confidence classes have orthologues across the three kingdoms (Fig. 1a and Supplementary Fig. 1). For example, 63%, 59% and 95% of the proteins in the Generic class have orthologues in eukaryotes, archaea and bacteria, respectively (Fig. 1a and Supplementary Fig. 1), rising to 91%, 70% and 99% for the Equivalog class. Only two proteins from the Unknown class had many orthologues in both eukaryotes and archaea. These proteins MMSYN1_0298' and MMSYN1_0302 were classified by Hutchison et al. into the Unclear and Cofactor transport and salvage functional categories, respectively. Our analysis determined confident functions for both of these proteins (see below).

### Domain architecture of minimal genome proteins. 
Domain analysis, using Pfam[10] (Supplementary Data File 2), showed that few (22%) of the proteins in the Unknown class contain known domains, significantly less than for the other four classes (Fig. 1b; $p < 8.3e{-}12$; Mann–Whitney–Wilcoxon test). In contrast, all proteins in the Equivalog class contain at least one domain and nearly half of them (44%) have a multi-domain architecture (Fig. 1b), whereas multiple domains are present in 21% of the proteins in the Generic class and only a single protein in the Unknown class (Fig. 1b). The proteins in the Unknown class are also clearly different to those in the Generic class, where a domain is present in 86% of the proteins. Further, the proteins in the Unknown class also have more disordered regions than the other groups (Fig. 1c), although this does not reach statistical significance (X-squared = 19.304, df = 16, $p = 0.2532$; Chi-Square test for categorical data).

### Structural modelling of the minimal genom. 
Hutchison et al.[1] used threading (an approach for modelling protein structure) to support functional assignment from TIGRFAM matches. Here, the Phyre2[11] protein structure prediction server was used to model the structures of the minimal genome proteins. With the exception of the Unknown class, high confidence structural templates were identified for the vast majority of proteins for at least part of the sequence (Supplementary Fig. 2 and Supplementary Data File 3). The proportion of proteins in each confidence class that could be accurately modelled was considered by identifying those for which at least 75% of the protein sequence could be modelled with a structural model confidence score (from Phyre2) of at least 90%. In the Unknown class this applied to only nine proteins, whereas nearly all proteins in the four other confidence groups were successfully modelled (Fig. 1d).

### Transmembrane proteins. 
Proteins in the Unknown and Generic classes are enriched with transmembrane proteins with 49% and 35%, respectively, of their proteins predicted to have transmembrane helices (Fig. 2a and Supplementary Data File 4). In contrast, very few transmembrane proteins were identified in the Equivalog and Probable classes (6% and 12% respectively), while 32% of the proteins in the Putative class are transmembrane proteins (Fig. 2a).

These results suggest that many of the proteins that have unassigned functions may be associated with membranes. For example, 24 proteins in the Generic class are predicted to contain six or more transmembrane helices (Fig. 2b), many of which are likely to be transporters of essential nutrients from the media (see below).

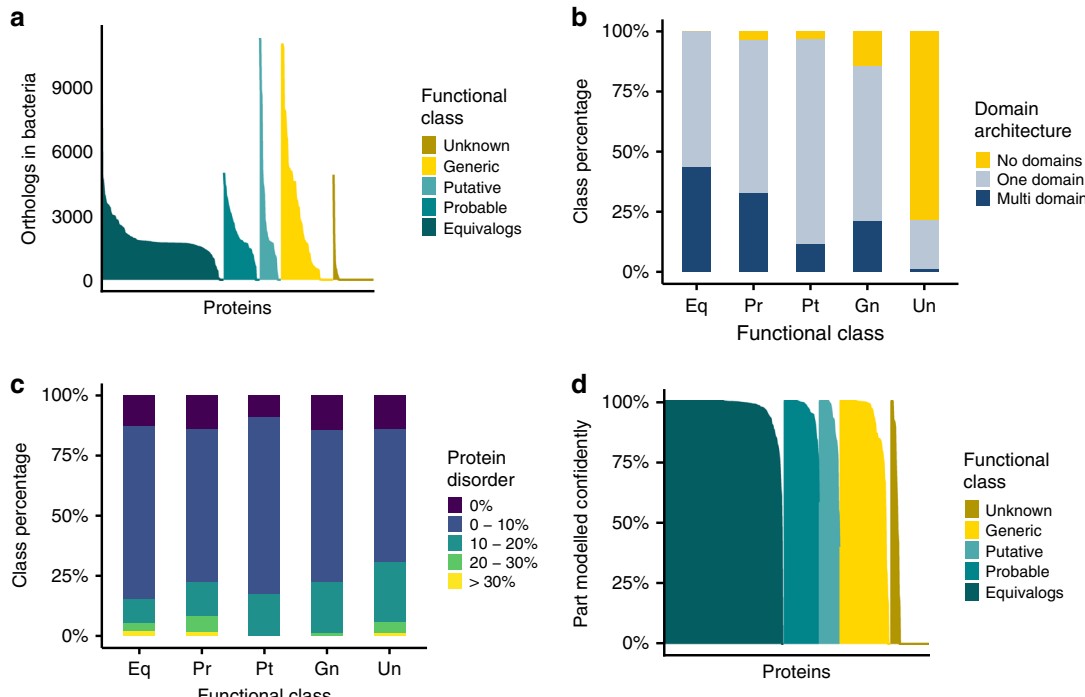

**Fig. 1** Basic characterisation of proteins encoded by the minimal bacterial genome. **a** Orthologues identified in bacteria. Results for each functional class are represented by a different colour: gold for the Unknown class, yellow–Generic, light turquoise–Putative, turquoise–Probable and dark turquoise–Equivalog. **b** The domain architecture for proteins in each of the five functional confidence classes is plotted (Unknown [Un], Generic [Gn], Putative [Pt], Probable [Pr] and Equivalog [Eq]). Proteins with no domains are displayed in yellow, grey represents single domain proteins and dark blue multi-domain proteins. **c** Predicted protein disorder in the minimal genome proteins. The results are shown for the five confidence classes from **b** and coloured according to the percentage of disorder present. Proteins with a percentage disorder >30% are represented by yellow, 20–30% disorder by green, 10–20%-turquoise and 0–10%-blue. Purple indicates proteins without disordered regions. **d** The percentage of protein structure that can be confidently modified by Phyre2. Functional class colouring as for **a**

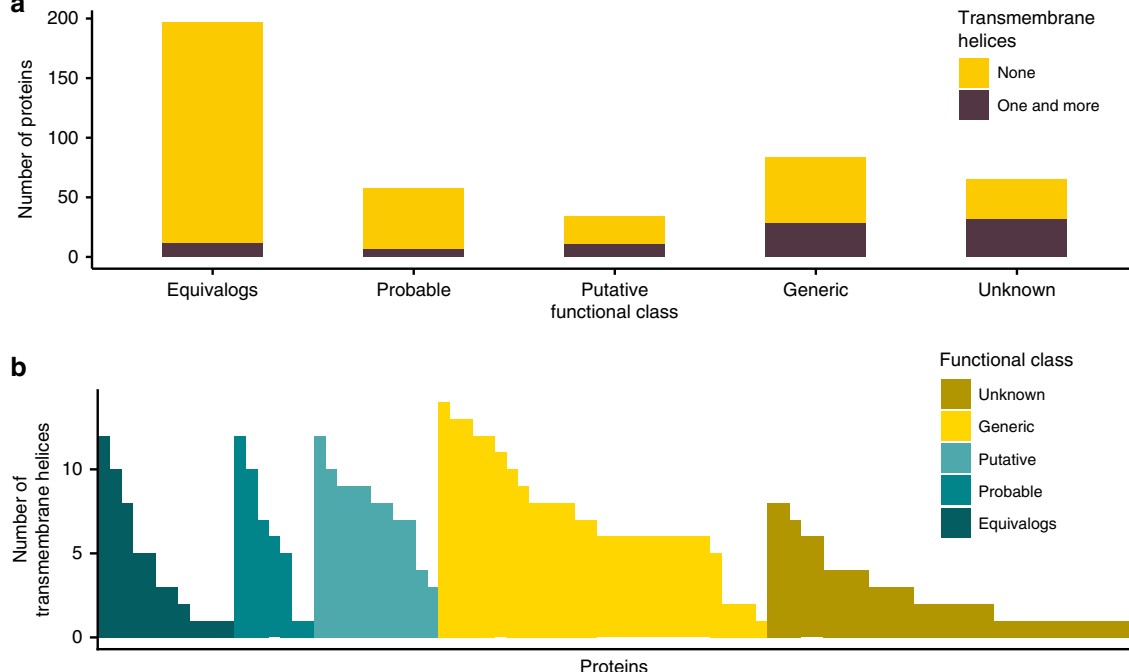

**Fig. 2** Transmembrane proteins encoded by the minimal bacterial genome. **a** The number of proteins predicted by TMHMM to have transmembrane helices. Brown indicates proteins with one or more transmembrane helix. Yellow for those without transmembrane helices. **b** The number of transmembrane helices present in each of the proteins in the minimal genome that is predicted to have one or more transmembrane helix. Results for each functional class are represented by a different colour: gold for the Unknown class, yellow–Generic, light turquoise–Putative, turquoise–Probable and dark turquoise–Equivalog class

**Prediction classification for specificity and confidence**. To infer functions for the proteins of unknown function, we introduced a different way to classify our results, which separates function specificity and prediction confidence. This enabled a more nuanced interpretation of the results than the five classes (Unknown to Equivalog) used by Hutchison et al., which combined both specificity and confidence. Our specificity classes include 'hypothetical', where the function is completely unknown, 'general', where we have some basic functional information (e.g., DNA binding or transporter), 'specific', where we have identified a specific function (e.g., transcription factor, ABC transporter) and 'highly specific', where a high level of detail is known (e.g., ABC transporter with known substrate; further examples are given in Supplementary Table 1).

We use the number of methods that support a function and the average score associated with this function as indicators of the confidence of the annotation (see methods). The average score for each predicted function was calculated by normalising the scores from the individual methods (e.g., e-value or probability) to the range of 0–100, with 100 indicating a highly confident score (e.g., a highly significant e-value from Pfam or Gene3D; see methods). Further, each protein was assigned to a larger functional category that represents biological process using the 30 different functional categories proposed by Hutchison et al.

Before predicting protein functions, we re-analysed the annotations by Hutchison et al. and assigned the functions to our new specificity classes. Confidence levels of these initial functional annotations could not be compared, since the outputs of the individual methods from the Hutchison et al. study were not available. Our assignment to specificity classes shows that most of the proteins in the Putative, Probable and Equivalog classes had previously been assigned highly specific functions, highlighting how the classification combined both functional specificity and confidence (Fig. 3a). Further, this analysis suggested that for some of the proteins classed as of unknown function (particularly the Generic class), there had been some suggestion of function, but with very low confidence (Fig. 3a), i.e., these were long shots based on the results from the three methods used in the Hutchison et al. study. Most of the proteins in the Unknown class were considered to be 'hypothetical' according to our criteria (Fig. 3a).

**Benchmarking our approach using proteins of known function**. In contrast to Hutchison et al.[1], who used TIGRFAM, genome context and threading to functionally characterise the proteins encoded by the minimal genome, we applied a wider range of approaches to infer their functions. Many methods have been developed to predict protein function using properties ranging from protein sequence to interaction data and predicting features ranging from subcellular localisation to Gene Ontology (GO) terms and protein structure[12]. Here, we applied the top performing methods from the recent CAFA[7,8] assessments, which were available as either a webserver or for download in combination with other established methods to assign functions to the proteins encoded by the minimal bacterial genome (see methods and Fig. 4). Overall functional inferences were made by manually investigating and combining the predictions and their consistency with genes from the same operon.

To test the performance of our approach, we applied it to the proteins of known function belonging to the Hutchison classes Putative, Probable and Equivalog. For 92% (266 of 289) of the proteins, the functions predicted by our approach agreed with the annotation assigned by Hutchison et al. (Fig. 3b). Our approach has increased the confidence of these annotations, with an average of 13 methods making predictions that supported the

functional annotations, compared to a maximum of three methods used in the previous study (Fig. 3c).

For nine proteins there were minimal differences in the annotations, for example MMSYN1_0637 was previously annotated as the gene rpsI, which encodes the 30S ribosomal protein S9, whereas our predictions suggest it to be rpsN, which encodes the 30S ribosomal protein S5 (Supplementary Data File 5), which is probably due to them both belonging to the ribosomal protein S5 domain 2-like superfamily. For 12 proteins, our annotations were less specific than the original ones. These proteins were solely in the Hutchison et al. Putative class and the existing annotations were highly specific (Supplementary Data File 5), such as for MMSYN1_0787, our annotation of RelA/SpoT family protein, is more general that than the original relA gene annotation. For a single protein (MMSYN1_0154) our predicted function of leucyl aminopeptidase was more specific than the initial cytosol aminopeptidase family, catalytic domain protein. Further, only for a single protein (MMSYN1_0908) was our predicted function (yidC; inner membrane protein translocase component) completely different to the existing annotation (misC-polyketide synthase). Overall, this demonstrates that for proteins with known function our approach is able to assign functions that agree with the existing annotations although in some cases, our assignment may be less specific than the existing annotations. We did not assign functions that disagreed with the known function. Further, with many methods now supporting these functions, there is greater confidence in them.

**Annotating proteins of previously unknown function**. We assigned a function to 133 of the 149 proteins of unknown function. For nearly half of them (66 of 149), new functional information was provided. This included more specific functions (25), assigning a functional category (5) or both of these (26). For the remaining ten proteins, greater functional information was added but the specificity class or functional category remained the same. For example, MMSYN1_0133 was initially annotated as a peptidase of the S8/S53 family, while we proposed a Subtilisin-like 1 serine protease function. While our annotation is more detailed, it is not highly specific and so the protein remained in the Specific class and Proteolysis functional category.

For 51 proteins, a more specific function was assigned (Fig. 5a and Supplementary Data File 5). For 33 proteins that had initially been annotated as hypothetical, a function was now assigned. Twenty-five of these annotations were classified as General, seven as Specific and one as Highly specific (Fig. 5a and Supplementary Data File 5). Eight proteins moved from a General to a Specific function (seven Specific, one Highly specific), and 10 proteins were assigned Highly specific functions having previously been assigned a Specific function (Fig. 5a). These predictions vary in their level of confidence. Some of them are supported by many methods, while some have highly confident predictions from a smaller number of methods (Fig. 5b, c).

For most proteins that were assigned a general function, we see that they were often supported by fewer methods but those methods predicted them with high confidence scores (Fig. 5a). For example, the group of proteins in the bottom right corner of Fig. 5c were all predicted to be transporters but only assigned a general function as further details such a substrate specificity could not be inferred. Where Specific and Highly specific functions were assigned, typically more methods supported the function but there was a greater range in the scores associated from the individual methods (Fig. 5). For example, MMSYN1_0298 and MMSYN1_0302 were both initially classed as hypothetical and we have assigned them Specific and Highly specific functions respectively, based on data available from 10

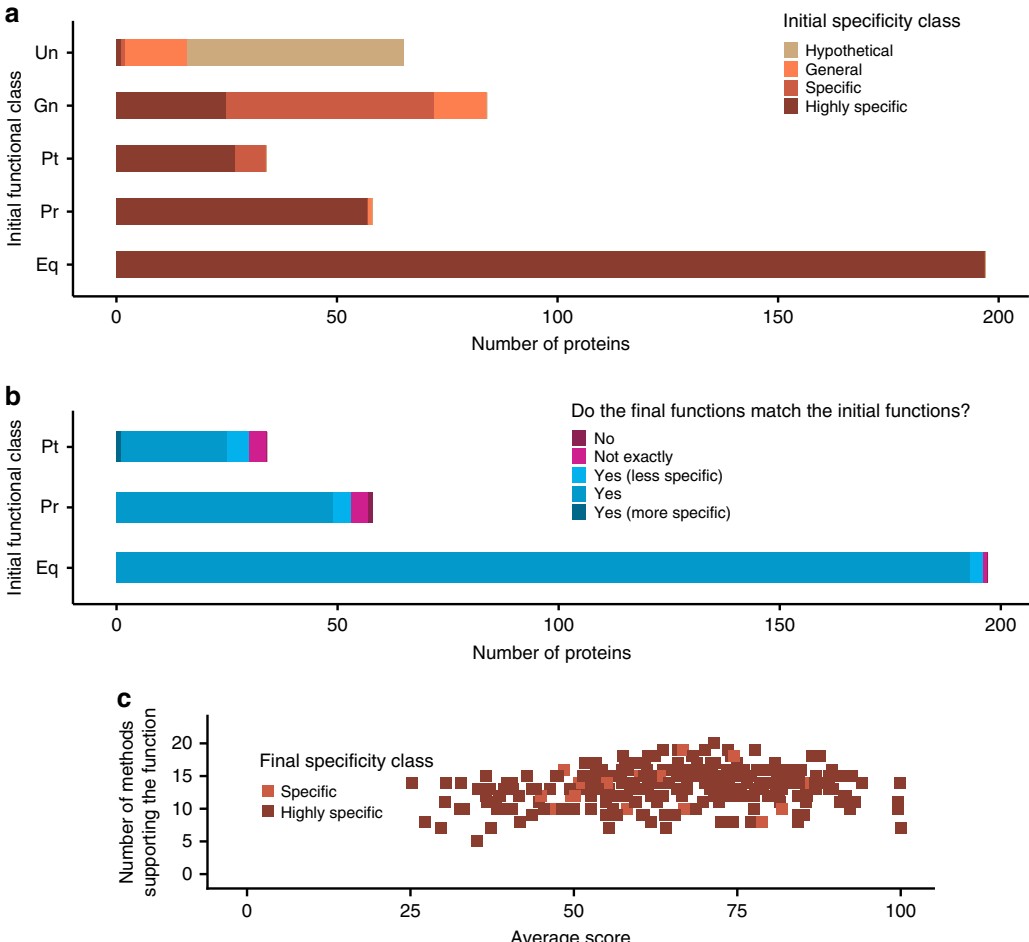

**Fig. 3** Predictions for proteins of known function encoded by the minimal genome. **a** Assessment of the specificity of functions predicted by Hutchison et al. across all five initial functional classes (Unknown to Equivalog). Functions from different initial specificity classes are represented by a different colour: beige for the Hypothetical specificity class, orange–General, light brown–Specific and dark brown–Highly specific. **b** Comparison of our predictions with the functions predicted by Hutchison et al. for proteins of known function, i.e., from the Putative, Probable and Equivalog functional classes. Colouring indicates the level of agreement between the initial functions and the predictions made here. Dark blue where the functions exactly match, medium blue where the predictions made here were less specific than the initial ones, light blue where our predictions were more specific, dark purple where there were minor differences between the functions and light purple where the function did not agree. **c** Number of methods supporting the function and the average score of those methods. Each point represents a protein. Methods include those used in the first step of function prediction. Specificity class colouring as for **a**

(MMSYN1_0298) and 12 (MMSYN1_0302) methods (Fig. 5 and Supplementary Data Files 1–6, 8). Based on these data sources we propose that MMSYN1_0298 is a ribosomal protein from the family L7AE/L30e (Fig. 6a) and that MMSYN1_0302 is an oxygen-insensitive NAD(P)H nitroreductase (Fig. 6b), both of which are functions widespread across the kingdoms of life.

Our analysis suggests that the combination of methods improves the reliability of function annotation. For some proteins, there appeared to be evidence for a given function from multiple sources, but on closer inspection it was difficult to assign more confident annotations (Supplementary Fig. 3). For example, MMSYN1_0138 is homologous to the ATP-binding region of ABC transporters but the ATP-binding site is not conserved, which casts some doubt on this function (Supplementary Fig. 3A). For MMSYN1_0615, matches from four methods suggest a Phenylalanine-tRNA ligase function (Supplementary Fig. 3B). However, MMSYN1_0615 only contains 202 residues and the beta chain of bacterial Phenylalanine-tRNA ligases contain nearly 800 residues, making it unlikely that MMSYN1_0615 performs this function (Supplementary Fig. 3B).

Overall, we found that the diversity of different methods used was required for inferring function, with no individual method

able to predict the most detailed function assigned to more than one-third of the proteins of unknown function (Supplementary Table 2). The top five methods to assign the most detailed functions each used different approaches, including a method that identifies orthologous groups (eggNOG-Mapper[9]), the group of methods that predict GO terms, a method that predicts protein three-dimensional structure (Phyre2[11]), identification of protein domains from Pfam and finally the best BLAST match from UniProt. Further, any combination of the top five performing methods only obtained the final annotation for a maximum of 25% of the proteins, further highlighting the contribution of multiple different methods to assign functions (Supplementary Table 3). Two methods (GO terms and TMHMM) were able to widely provide more generic functions supporting the overall assigned function (54% for GO terms and 82% for TMHMM), although TMHMM only predicts if the protein contains transmembrane helices (Supplementary Table 2).

For the remaining 83 proteins, our predictions supported the existing annotation. Importantly for many of these proteins, multiple methods have now made predictions that support the annotation, thus increasing their confidence. Figure 7 shows that many of the proteins (28 out of 83) have predicted functions that

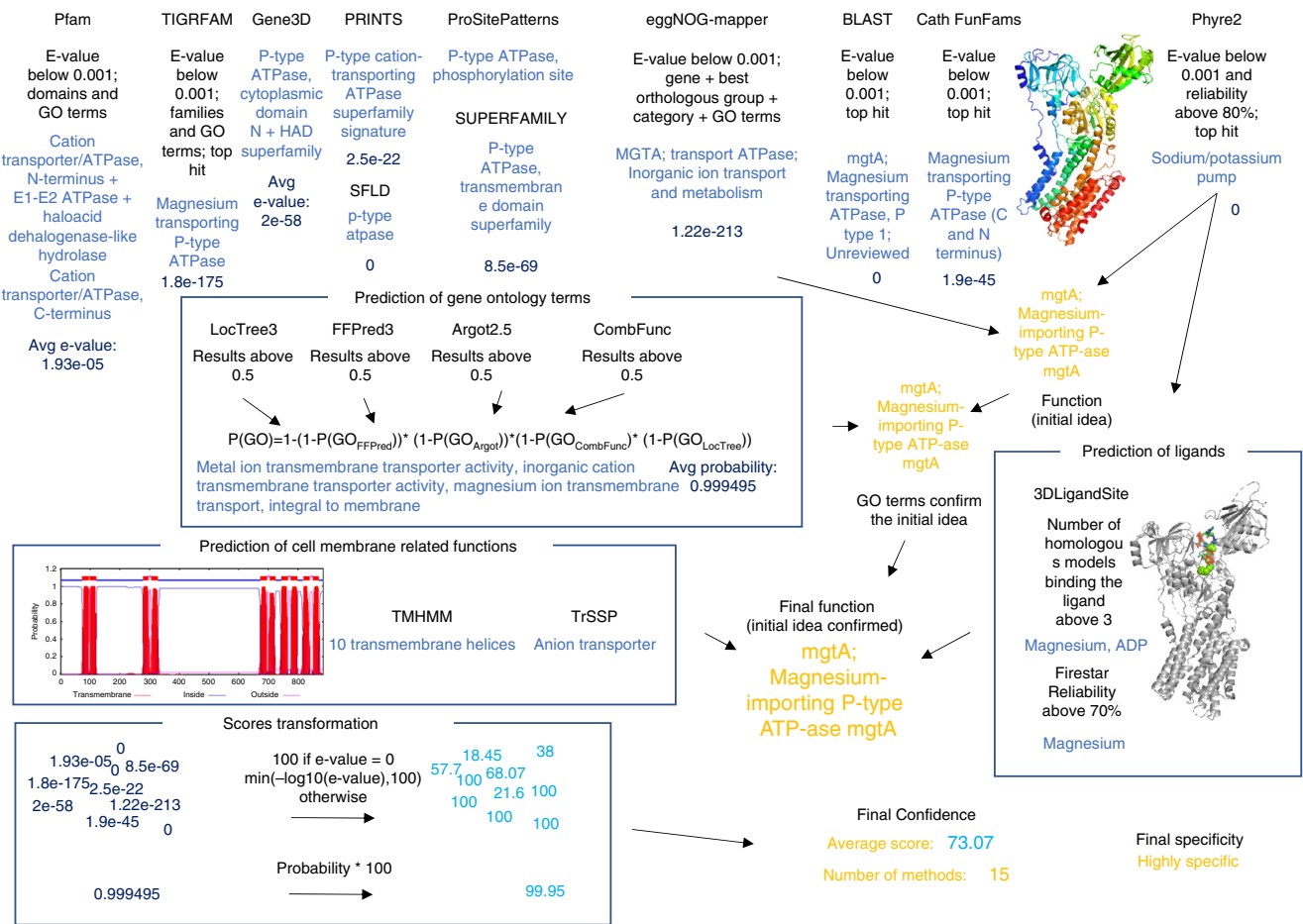

**Fig. 4** Assigning function to proteins in the minimal genome. The flowchart outlines how functions were assigned to the proteins using MMSYN1_0879 as an example. The top row of methods are used to identify a likely function. The methods in the three groups of boxes (predicted GO terms, ligand binding predictions and membrane protein predictions) are then used to see if they support the function identified by the first group. Where the first group does not predict a function then this second group was used. The figure shows the results obtained for MMSYN1_0879, which was annotated as the gene mgtA, a magnesium importing P-type ATPase

are supported by 10 or more methods, rising to 61 supported by 5 or more methods, often with high confidence scores (or *e*-values) from the individual methods.

**Understanding biological processes in the minimal genome.** Functional categories were assigned to 31 proteins that had previously been classified with Unclear biological process. The majority of the proteins with a newly assigned functional category were predicted to have transporter functions, with 24 proteins added to the 84 already assigned to this functional category (Fig. 8a). Further, one protein (MMSYN1_0033) was assigned to the cytosolic metabolism category, three to the preservation of genetic information category (MMSYN1_0005, MMSYN1_0239, MMSYN1_0353), and three to the expression of genetic information category (MMSYN1_0615, MMSYN1_0730, MMSYN1_0873) (Fig. 8a).

Overall, while functional annotations have been inferred for a considerable proportion of the proteins of unknown function, the biological process for 48 proteins remains unknown (i.e., in the Unclear category; Fig. 8a). For 32 of these proteins, a molecular function was assigned such as Cof-like hydrolase, ATPase AAA family, or DNA-binding protein HU, but there was insufficient information to assign a functional category. The remaining sixteen proteins lack functional information and are classified as hypothetical. These proteins do not contain any known domains or transmembrane helices, none have orthologues in other kingdoms of life and only a few within bacteria. Either these are species-specific proteins that perform an important function within Mycobacteria or they have diverged significantly such that sequence relationships are not detected.

**Newly assigned functions indicate transporters.** Transmembrane helices were identified in 41% (61) of the proteins of unknown function (Fig. 2 and Supplementary Data File 4). Fifteen transmembrane proteins, which were not categorised as transporters, were annotated with functions in cell division (1), chromosome segregation (1) and proteolysis (4), while the biological process remained unknown for nine. Our analysis suggests that 46 of the 61 predicted transmembrane proteins are likely to be responsible for membrane transport (Supplementary Data File 4, S6). Of the 46, 23 were previously annotated by Hutchison et al. with a range of transporter functions (e.g., ABC transporters, S component of ECF transporters), all of which were further supported by our analysis. A further 15 proteins that lack transmembrane domains were also associated with transport functions, e.g., ATP-binding units of ABC transporters, 14 of them were identified by Hutchison et al.[1].

Of the 24 newly proposed transporters (previously hypothetical or with minimal information, e.g., membrane protein), six gained specific transporter functions. All six were previously classed as

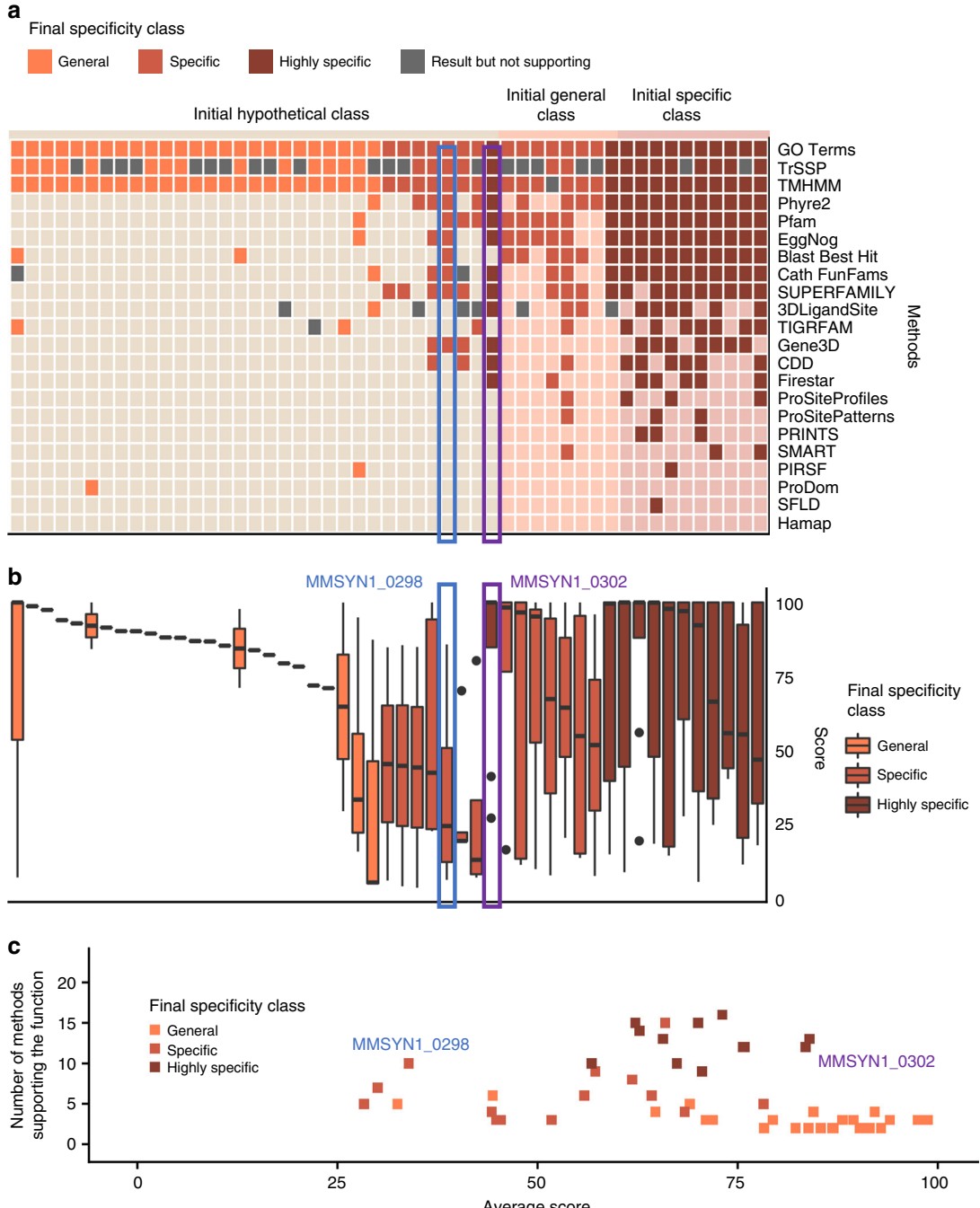

**Fig. 5** Proteins assigned new functions. This figure shows the 51 proteins where the specificity class was increased. Results for each final specificity class are represented by a different colour: orange for the General specificity class, light brown–Specific and dark brown–Highly specific. **a** Each column represents a protein in the minimal genome and the squares show the methods that made predictions (darker colours indicate support of the final prediction), grey squares indicate predictions that did not support the function, light squares indicate that a method did not make a prediction. Proteins are grouped by their initial specificity class (Hypothetical, General, Specific and Highly specific) and then by their final specificity class. **b** Boxplot demonstrating the distribution of the scores across proteins. Proteins grouped by their initial specificity class and then by their final specificity class. Horizontal lines represent the median, the lower and upper hinge show respectively first quartile and third quartile, and lower and upper whisker include scores from first quartile to (distance between the first and third quartile) × 1.5 (for lower whisker) and from third quartile to (distance between the first and third quartile) × 1.5 (for upper whisker). Any scores outside of these intervals are shown as points. **c** The number of methods supporting the function and the average score. Each point represents a protein

membrane proteins and have now been annotated as transporters; one hexose phosphate transport protein (MMSYN1_0881), one ABC transporter (MMSYN1_0411), one S component of an ECF transporter (MMSYN1_0877), and three belonging to the Major facilitator superfamily (MMSYN1_0235, MMSYN1_0325,

MMSYN1_0478) (Supplementary Data Files 1–6, 8). The remaining 18 proteins annotated as transporters (general specificity level) had previously either been annotated as membrane or hypothetical proteins. Results from a few methods (with high scores–Fig. 9) indicate that they are transporters but it

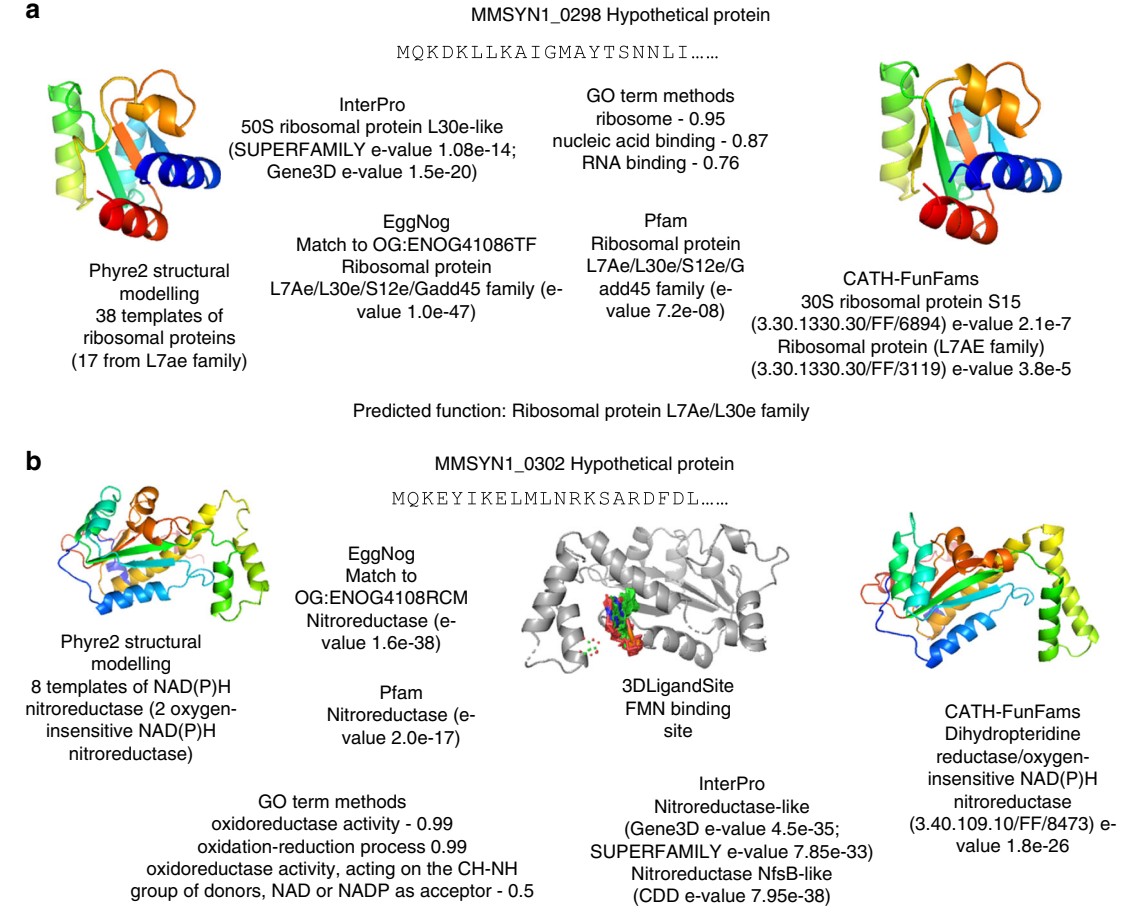

**a** MMSYN1_0298 Hypothetical protein

MQKDKLLKAIGMAYTSNNLI……

InterPro
50S ribosomal protein L30e-like
(SUPERFAMILY e-value 1.08e-14;
Gene3D e-value 1.5e-20)

GO term methods
ribosome - 0.95
nucleic acid binding - 0.87
RNA binding - 0.76

Phyre2 structural
modelling
38 templates of
ribosomal proteins
(17 from L7ae family)

EggNog
Match to OG:ENOG41086TF
Ribosomal protein
L7Ae/L30e/S12e/Gadd45 family (e-
value 1.0e-47)

Pfam
Ribosomal protein
L7Ae/L30e/S12e/G
add45 family (e-
value 7.2e-08)

CATH-FunFams
30S ribosomal protein S15
(3.30.1330.30/FF/6894) e-value 2.1e-7
Ribosomal protein (L7AE family)
(3.30.1330.30/FF/3119) e-value 3.8e-5

Predicted function: Ribosomal protein L7Ae/L30e family

**b** MMSYN1_0302 Hypothetical protein

MQKEYIKELMLNRKSARDFDL……

EggNog
Match to
OG:ENOG4108RCM
Nitroreductase (e-
value 1.6e-38)

Pfam
Nitroreductase (e-
value 2.0e-17)

Phyre2 structural
modelling
8 templates of NAD(P)H
nitroreductase (2 oxygen-
insensitive NAD(P)H
nitroreductase)

3DLigandSite
FMN binding
site

CATH-FunFams
Dihydropteridine
reductase/oxygen-
insensitive NAD(P)H
nitroreductase
(3.40.109.10/FF/8473) e-
value 1.8e-26

GO term methods
oxidoreductase activity - 0.99
oxidation-reduction process 0.99
oxidoreductase activity, acting on the CH-NH
group of donors, NAD or NADP as acceptor - 0.5

InterPro
Nitroreductase-like
(Gene3D e-value 4.5e-35;
SUPERFAMILY e-value 7.85e-33)
Nitroreductase NfsB-like
(CDD e-value 7.95e-38)

Predicted function: Oxygen-insensitive NAD(P)H nitroreductase

**Fig. 6** Confident predictions of protein function in the minimal genome. Both **a** MMSYN1_0298 and **b** MMSYN1_0302 were previously classified as hypothetical proteins. The results from prediction methods and the function assigned are shown

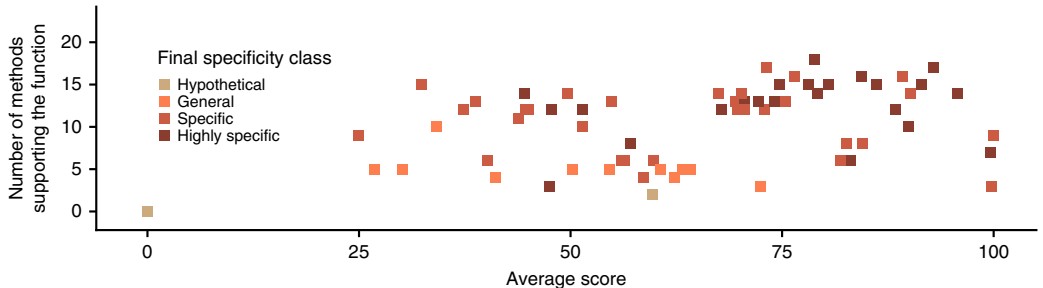

**Fig. 7** Multiple methods supporting existing annotations. For all proteins where the predicted function agreed with the existing annotation (i.e., the specificity class was not changed), the number of methods that predicted the function is plotted against the average score from these methods. Points for each of the final specificity classes are represented by a different colour: beige for the Hypothetical specificity class, orange–General, light brown–Specific and dark brown–Highly specific

was not possible to assign them to a specific family/type of transporter or to identify a substrate.

More specific annotations could be made for proteins already annotated with transport-related functions, including four proteins (MMSYN1_0034, MMSYN1_0399, MMSYN1_0531, MMSYN1_0639) that were classed as FtsX-like permeases having previously been given generic transport-related annotations (e.g.,

permease). For most of these proteins, we have greater confidence in the assigned function, given that many different methods support them (Fig. 9). This extends their initial annotations that had been assigned by only three methods. For example, one operon encodes proteins that transport oligopeptides (AmiABCDE MMSYN1_0165 - MMSYN1_0169) and another operon encodes a spermidine/putrescine transporter (PotABCD

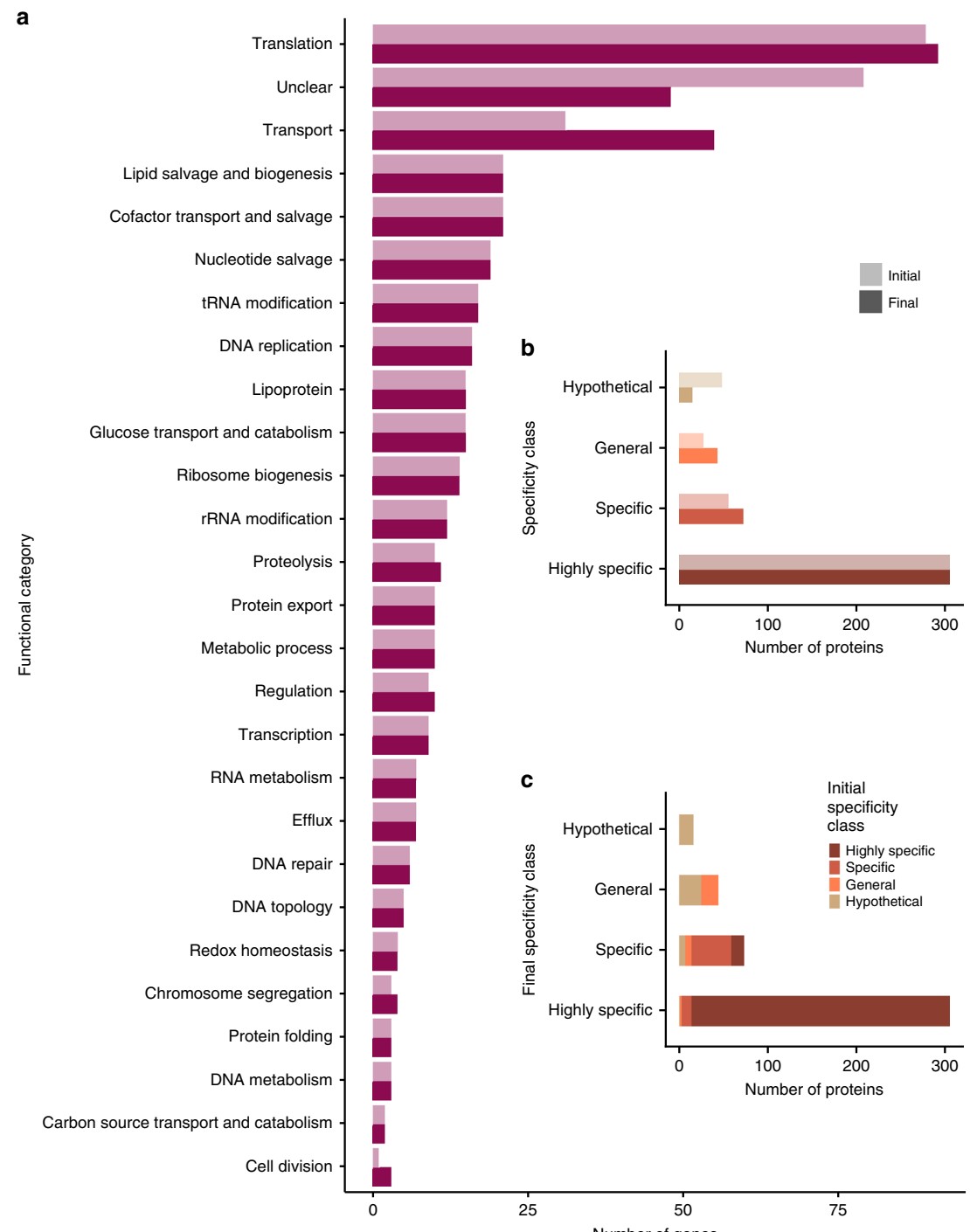

**Fig. 8** Functional annotations of the minimal bacterial genome. The number of proteins in each of the **a** protein biological process categories (light and dark purple indicate initial and final categories, respectively). **b** Specificity classes is shown with the original minimal genome annotation and the annotations identified here. **c** Shows the change in specificity classes, coloured based on the original specificity class. Results for each initial specificity class are represented by a different colour: beige for the Hypothetical specificity class, orange–General, light brown–Specific and dark brown–Highly specific

MMSYN1_0195 - MMSYN1_0197) (Supplementary Data File 5 and Supplementary Figs. 4 and 5).

One of the three proteins newly proposed to be members of the Major facilitator superfamily, MMSYN1_0325, was previously classified as a membrane protein (Fig. 10). In agreement, the transmembrane helix prediction tool TMHMM[13] predicted 13 transmembrane helices in the protein. Further, the structure was confidently modelled by Phyre2, with > 98% confidence for 26

independent structural templates, all of which had transporter functions (including members of the MSF superfamily). Inter-Pro[14] assigned it into the MFS transporter superfamily. Supporting this function, further methods predicted a range of transporter-related functions, including symporter activity (GO:0015293) and substrate-specific transmembrane transporter activity (GO:0022891) with probabilities >90% (Fig. 10 and Supplementary Data File 6).

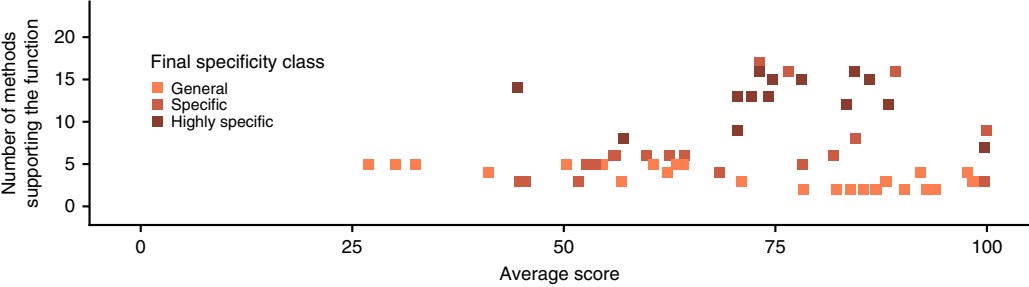

**Fig. 9** Prediction of membrane related functions. Each point represents a protein with initially unknown function for which we assigned cell membrane related functions (e.g., transmembrane, transporter). The number of methods that supported the prediction is plotted against the average score from these methods. Points for each of the final specificity classes are represented by a different colour: orange for the General specificity class, light brown–Specific and dark brown–Highly specific

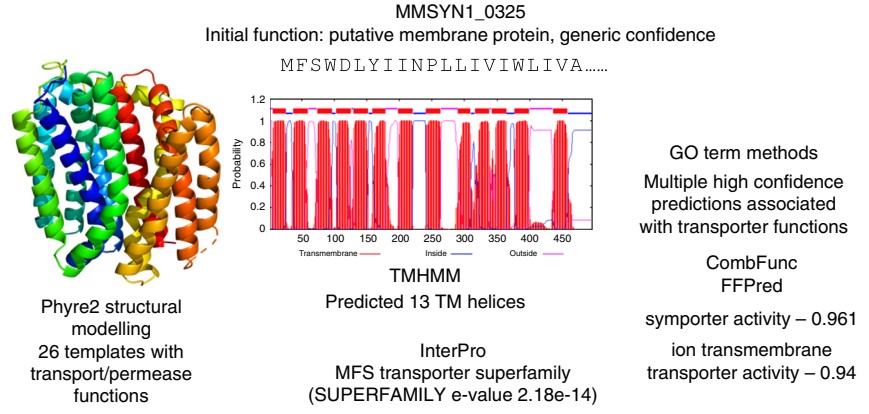

**Fig. 10** MMSYN1_0325 is predicted to be a transporter and member of the Major facilitator Superfamily. The results from Phyre2, TMHMM, the combination of GO term prediction methods (numbers shown are probability associated with each function) and InterPro are shown. All of these methods supported a transporter function with Phyre2 and InterPro confidently identifying association with the Major facilitator superfamily

**Comparison of predictions made by Danchin and Fang.** Recently Danchin and Fang[15] used what they referred to as an engineering-based approach to investigate the unknown functions within the minimal bacterial genome and provided annotations for 71 of the 149 proteins of unknown function. They set out to identify functions that would be expected to be in a minimal genome but were missing from the existing annotation and to then identify proteins that could perform these functions (although it is not clear how these candidates were identified as no methods were provided[15]).

Comparison of the results from both studies revealed considerable overlaps (Supplementary Data File 7). Using our approach, only sixteen proteins remained hypothetical without any assigned function, while Danchin and Fang did not provide any annotations for 78 of the proteins with unknown function. Thus, we leave only 10% of the previously unannotated proteins without any assigned function, while 52% remain completely uncharacterised by Danchin and Fang. This demonstrates the breadth of function that our approach is able to assign. The predictions showed complete agreement for 36 proteins and minor differences for 18 proteins (Supplementary Data File 7). For a further 13 proteins the predictions were more detailed in one study than the other (Supplementary Data File 7). For example, Danchin and Fang proposed that MMSYN1_0822, is an S component of an ECF transporter and is part of a folate transporter, whereas we identified three possible folate

transporters (MMSYN1_0314, MMSYN1_0822, MMSYN1_0836) and could not confidently assign substrates to any of them.

Four of the predictions differed considerably (Supplementary Data File 7). They are represented by proteins such as MMSYN1_0388 which here was annotated as a transmembrane protein, possibly a cation transporter, while Danchin and Fang suggested that it has a role in double-strand break repair. For three of the proteins, Danchin and Fang inferred more functional characteristics. They annotated MMSYN1_0853 MMSYN1_0530, MMSYN1_0511 with the functions energy-sensing regulator of translation, promiscuous phosphatase and double-strand break repair protein, respectively, while here they were retained as hypothetical since there was little agreement between the multiple methods used to be able to infer protein function.

**Discussion**

The synthesis of the bacterium with the smallest genome (to date), resulted in an astounding number (149 of 473) of proteins of unknown function and emphasised the gaps in our understanding of the basic principles of life. Our results demonstrate that the combined use of a range of complementary advanced methods for protein function inference is superior to the use of individual approaches. Using a combination of results from 22 different methods, we were able to assign new functional information to 66 of the 149 proteins that were originally classed as having unknown function. Further, given the use of many

different methods, we have increased the confidence in existing annotations that our approach also supported. For some proteins, more detailed functions were predicted by some of the methods. However, in manually combining the predictions, there was insufficient evidence to assign them to more specific functional classes. Nevertheless, these functions should be sufficient to direct further research and experimental characterisation. Our analysis shows that the combination of many methods was essential with no single method able to identify the highest detailed function assigned to more than one-third of the proteins (Supplementary Table 2).

Most of the proteins of unknown function were homologous to few other proteins with known functions and they also lacked orthologues. Thus, for many of the proteins where functions have been assigned, methods that are not dependent on homology were prevalent (e.g., FFPred3[16], Fig. 5 and S6). This highlights the importance of developing further methods that do not rely on homology. Moreover, many of the difficult to characterise proteins do not contain known protein domains and are enriched for transmembrane proteins (Fig. 1). Hence, additional approaches to predict the function of such proteins are required.

With our expanded functional assignments, 50% of the proteins encoded by the minimal genome perform functions associated with two fundamental life processes; preserving and expressing genetic information (Fig. 8a). Most notably, many proteins were assigned transporter functions, and these proteins now represent 22% of the minimal genome. In generating the minimal genome, 32 M. mycoides genes with membrane transport functions were removed[1]. Additionally, many proteins with metabolic functions were removed. Hence, the minimal genome bacterium is reliant on obtaining many nutrients from the medium and also needs to remove (toxic) metabolites from the cell. Thus, it may not be surprising that transporters are essential for the bacterium. It was not possible to assign substrates for these transporters. The reason for this may be at least in part due to the promiscuity of myco-plasmal transport systems[17]. Additionally, transporters may transport low-affinity substrates in a nutrient-rich environment in which nutrients are highly abundant.

The identification of many transporters also highlights the dependence of the minimal bacterial genome cells on the medium in which they grow. Hence, the nature of the minimal genome is partly shaped by the conditions in a nutrient-rich environment. Consequently, we propose that a minimal genome consists of two sets of genes. The first set encodes functions that are an essential prerequisite for all bacteria and probably all forms of life, which on its own is not sufficient to enable life. This gene set needs to be complemented with an additional set of genes that enables life in a particular environment. In a nutrient-rich environment, these additional genes may largely have functions associated with compound uptake and efflux in agreement with our current results presented here. Under other circumstances, where nutrients are not so abundant, metabolic functions are likely to be of greater importance.

In summary, we have successfully applied a combined bioin-formatics approach to characterise proteins with unknown function from the minimal genome that had not been annotated by previous approaches. Currently, only about 1% of all known proteins are annotated with experimentally confirmed functions. Since the experimental analysis of protein function will for the foreseeable future remain restricted to a small subset of proteins due to physical and financial limitations, optimised bioinfor-matics approaches will be critical for the assignment of functions to proteins and, in turn, our understanding of the essential functions of life. Proteins that are difficult to classify typically (i) do not contain known protein domains (ii) lack homology to proteins with known structure and (iii) are enriched for

transmembrane proteins. Further, most of the hypothetical pro-teins appear to be bacteria- and clade-specific. Hence, further complementary approaches are needed that enable the assign-ment of functions to such proteins. Importantly, a considerable proportion of the newly annotated proteins probably have transporter functions. These transporters are likely to be involved in the uptake of nutrients and efflux of waste products in a minimal genome organism that lacks many metabolic enzymes and is cultivated in a nutrient-rich environment. Additionally, our findings indicate the existence of a core set of genes that is essential for all forms of life but not sufficient to enable life on its own. This essential gene set needs to be complemented by a second enabling gene set that facilitates life under particular environmental conditions. Thus, the concept of a minimal gen-ome is context/environment specific.

## Methods

**Identifying basic protein properties**. Protein domains were determined by running PfamScan against the library of Pfam 30.0 HMMs[10]. GO terms associated with Pfam domains were extracted using the pfam2go file[10] (version 11 February 2017). The e-value of the domain matches were used to indicate the confidence of a GO term describing the function of the query protein. To test if the probability of minimal genome proteins having more domains identified increases with the increasing confidence of the annotation in the particular functional class, we performed the Mann–Whitney–Wilcoxon test. We cross-compared all the functional classes (438 proteins in total) and tested the null hypothesis that samples have the same dis-tribution against the alternative hypothesis that there is a >0 shift in the distribution.

InterProScan was run with default settings to determine matches against InterPro databases of protein signatures[14]. Results from the following resources were included in the analysis: CDD[18], Gene3D[19], HAMAP[20], PIRSF[21], PRINTS[22] ProDom[23], ProSitePatterns[24], ProSiteProfiles[24], SFLD[25], SMART[26] and SUPERFAMILY[27].

Orthologues were identified using eggNOG-Mapper[9] against HMM databases for the three kingdoms of life. Additionally, precision of predictions was prioritised by restricting results to only one-to-one orthologues. The eggNOG-Mapper API was used to predict the orthologous groups in eggNOG that the minimal genome proteins belonged to. The proteins present in these orthologous groups were extracted and the species associated with the sequences were mapped to the NCBI Taxonomy to group them into phyla and used to identify the phyla where orthologues were present. Predicted features including GO terms, KEGG pathways and functional categories of Cluster of Orthologous Groups were also obtained from eggNOG-Mapper.

**Identifying membrane transporters and lipoproteins**. Proteins were classified as lipoproteins (SPaseI-cleaved proteins), SPaseI-cleaved proteins, cytoplasmic and transmembrane proteins using LipoP[28]. Similarly, proteins were distinguished between membrane transporters and non-transporters using TrSSP[29]. TrSSP pre-dicted substrates of the proteins from seven groups: amino acid, anion, cation, electron, protein/mRNA, sugar and other. The functions of membrane transporters and lipoproteins were further supported by identifying transmembrane helices, signal peptides and protein topology using TMHMM[13].

**Inferring gene ontology-based protein function**. GO terms were predicted using FFPred3[16], Argot2.5[30], CombFunc[31,32] (only Molecular Function terms) and LocTree3[33] (only Cellular Component terms). As the FFPred3 SVMs were trained only on human proteins from UniProtKB, predicted GO terms were additionally filtered using the frequency of terms in UniProtKB-GOA (version 5 June 2017). Predicted GO terms that were not annotated to any bacterial proteins in UniProtKB-GOA were removed from the set of FFPred3 predicted functions as they were likely to be functions that are not present in prokaryotes.

Argot2.5 was run with the taxonomic constraints option. As scores returned by Argot2.5 have a minimum score of zero and no upper bound, the linear spline function recommended by the method developers (personal communication) was applied to rescale them to the range of 0 to 1. CombFunc[31] was run using standard settings.

**Structural analysis**. The CATH FunFHMMer webserver was used to identify the functional families of structural domains, CATH FunFams[34,35].

Protein disorder was predicted using DISOPRED3[36]. For each of the proteins, the percentage of disordered regions was calculated based on the DISOPRED3 results. To verify if there is a statistically significant difference between 438 minimal genome proteins in five different functional classes, we performed a Chi-Square test for categorical data with a null hypothesis that the functional class of a protein and its disorder ratio level (0%, (0%, 10%], (10%, 20%], (20%, 30%], >30%) are independent.

Firestar[37] and 3DLigandSite[38] were used to predict ligands binding to the proteins. For Firestar only results marked as cognate were considered. Phyre2[11]

was run using standard mode to model the structure of the minimal genome proteins. Information provided by the name and description of the best matching models was used in the process of inferring function of the proteins. To make sure that each residue was covered with the highest possible confidence, the matches were firstly sorted by e-value and then selected gradually if they covered residues that were not covered before by a match with lower e-value.

**Identifying operons.** Genes in the synthetic *M. mycoides* (JCVI-syn1.0) were grouped into operons based on the predictions made for both *M. mycoides subsp capri LC* str 95010 and *M. mycoides* subsp mycoides SC str PG1 by two methods DOOR2[39] and MicrobesOnline[40]. The proteins of the synthetic *M. mycoides* were first mapped to the proteins of *M. mycoides subsp capri* LC str 95010 and *M. mycoides* subsp mycoides SC str PG1 downloaded from GenBank[41]. This was done by using BLAST to search against databases constructed from proteomes of these two species and extracting the best hit. A protein from *M. mycoides* subsp capri LC str 95010 or *M. mycoides* subsp mycoides SC str PG1 was considered a corresponding homologue of a protein from synthetic *M. mycoides* if the coverage and identity were greater than or equal to 80%. Via the corresponding homologues, operons predicted for these two species by DOOR2 and MicrobesOnline were mapped to the proteins of the synthetic *M. mycoides*.

**Combined protein function prediction.** The results from the following methods were removed from the analysis if their e-value was above 0.001: TIRGFAM, Pfam, eggNOG-Mapper, CATH FunFams and domains. Models predicted by Phyre2 were kept if the probability of the match was above 80% and e-value was below 0.001. Only results from Firestar with a reliability score above 70% and marked as cognate were retained. Ligands predicted by 3DLigandSite were kept if they were included in at least three homologous models. The best BLAST hit from UniProt (maximum e-value of 0.001) was used to identify the closest homologue of the protein and the information accessible in UniProt was taken into account in the annotation. Additionally, all the predictions of Gene Ontology terms were combined together and the probability of particular terms being predicted by any of the methods were calculated using the following formula: $P(GO) = 1 - (1 - P(GO_{FFPred3}))^* (1 - P(GO_{Argot2.5}))^* (1 - P(GO_{CombFunc})^* (1 - P(GO_{LocTree3}))$, where P(GO) is the combined probability of a given GO term and where subscripts are included this indicates the probability of that term from the named individual method. Only high probability ($> 0.65$) Gene Ontology terms were considered for each of the proteins. For the final prediction of protein function, results from all the methods were manually reviewed. The initial proposition of protein function was based on combining the results from TIGRFAM equivalog families, Pfam domains, InterPro families and domains, eggNOG orthologous groups, CATH functional families, the best BLAST hit from UniProt and the Phyre2 model of the structure. In considering the results from these methods, we looked for agreement between methods, particularly with highly confident results. This initial function was then verified using the predicted Gene Ontology terms and information on predicted ligands (Firestar[37], 3DLigandSite[38]) and transmembrane helices (TMHMM). Where information was not available from the first group of methods, the second group of methods were used as a starting point to infer functions. Transporters and lipoproteins were predicted using membrane transporter (TrSSP) and lipoprotein signal sequences (LipoP) respectively. Finally, it was considered if the predicted function was consistent within a group of genes in the same operon. Where methods made predictions that conflicted with the final predicted function, this was noted, but it did not affect the confidence as we recorded the number of methods supporting a function and the average score associated with these predictions (see below).

Confidence of predicted functions was considered for each protein by counting the results that support the final function and calculating the average score from these methods. Results used to calculate the average score come from the methods applied in the first step of function prediction (Fig. 4), i.e., TIGRFAM, Pfam, InterPro resources (all but ProSitePatterns), eggNog-Mapper, BLAST, CATH FunFams, Phyre2, and also the overall GO term-based prediction (which already combined Argot2.5, CombFunc, FFPred3 and LocTree3) resulting in 17 methods in total. Methods that concern a very specific element of a function, such as transmembrane helices or ligands were not included in the average score calculation. For all methods, scores were normalised to the range 0–100. Most of the methods use e-values as a measure of confidence (e.g., TIGRFAM, Pfam), for these methods –log10(e-value) was used capping the value at 100. Where probabilities were provided these were multiplied by 100. HAMAP and ProSiteProfiles use scores that are not probabilities or e-values and do not appear to have an upper bound. Considering the scores of these methods for the proteins of known function in the minimal genome indicated that scores were typically in the range 0–100 (Supplementary Fig. 7 and Supplementary Fig. 8), so scores above 100 were capped at 100.

**Reporting summary.** Further information on research design is available in the Nature Research Reporting Summary linked to this article.

## Data availability

The protein sequences encoded by the minimal genome were obtained from the supplementary material of Hutchison et al.[1]. A processed form of the full results provided by each of the methods used is provided in the supplementary data files. The raw results from the different methods are available from Figshare under https://doi.org/10.6084/m9.figshare.8218937

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

## Acknowledgements

We would like to thank Christopher Mulligan and Jose Ortega Roldan for helpful discussion about transporters, particularly ABC transporters and Prof. Stefano Toppo for advice regarding transformation of Argot2.5 scores into probabilities.

## Author contributions

All authors devised the study. M.A. performed the experiments. All authors analysed the results and wrote the paper.

## Additional information

**Competing interests:** The authors declare no competing interests.

