## [Peer Review File · Nature Communications]

Reviewers' Comments:

Reviewer #1:

Remarks to the Author:

I have read the manuscript by Wass and coworkers expecting a new methodology or insight in how to assign functions to proteins of unknown function. I must confess I am a bit disappointed. The authors use diverse methodologies and assign generic functions like transport to the majority of proteins and only assign a true functionality to a few. Moreover, there is no validation of the predictions at all, no bootstrapping, benchmark etc..., therefore Without some objective validation I cannot recommend publication of this work in Nature communications.

Reviewer #2:

Remarks to the Author:

This paper describes the application of a combined automatic and manual protocol for assigning protein functions. The approach exploits multiple computational methods and is used to functionally annotate a poorly characterised minimal bacterial genome – less than 1% of the proteins in this genome have experimental characterisation. Previous studies had predicted functions for proteins in this genome, to a reasonable level of confidence, for all but 149 of the 373 proteins. Sixteen different computational methods are applied, ranging from those identifying general features e.g. presence of transmembrane helices to those modelling the structure and predicting the function.

Nearly two thirds of these 149 uncharacterised proteins are assigned a putative function by this integrated computational pipeline with a significant portion being transporter functions most likely reflecting the need for nutrient uptake and waste disposal.

It is a thorough and comprehensive piece of work and the tabulated results represent a significant body of appropriate, rigorous analysis. The methodology is well designed with state-of-the-art resources/algorithms being applied to characterise the proteins in different ways. For each analysis, all the proteins in the minimal genome are analysed and the presentation of these results helps in understanding the extent to which the unknown category differs from the rest. It's interesting that such a high proportion of the proteins in the more confidently assigned functional class can be structurally modelled but so few of the unknown functional class, possibly because they are enriched in membrane proteins. This enrichment in membrane proteins certainly ties in with subsequent analyses suggesting these proteins are transporters.

Points to address

1. I found it difficult to understand what determined how an assignment changed category. In this context some of the statements made are very subjective and unclear. For example;
>For 59 proteins the confidence class was increased. 23 proteins >were assigned additional functional annotations, while for the >remaining 36 proteins the confidence in the existing annotation >was enhanced.

It's not clear to me what determines how confidence is 'increased' or just 'enhanced' and for the 23 that had additional functions if these are different functions does that mean confidence is decreased in the previous functions?

In this context, it would be helpful to have more rigorous definitions of what criteria need to be met to assign a protein from unknown to putative etc. On page 21 the statement

>If a specific function including a substrate and a biological >role was determined and confirmed by multiple sources of >evidence, a protein was classified in the putative confidence >class.

Similarly, if a general function was predicted and >supported by several lines of evidence, a protein was >classified in the generic functional class.

What is the quantitative difference between multiple and several?

In the sentence below – I'm not sure I understand what is meant by functional category being assigned if 'indicated coherently'.

Does that mean within some threshold set by the method?

For the GO terms, the authors calculate a combined probability value. I may have missed this, but as far as I could see there were no thresholds on this combined probability value, for assigning the GO term.

It would be helpful if the authors could give a clearer indication of how they used these probability values, presumably in their final manual assessment. The supplementary tables do give an excellent summary of the assignments of each method and are a very useful resource for biologists to assess the likely function and to guide experimentation.

2. Regarding conflicts in functional assignments, the authors state on page 21 that if one method points to one category eg transport and the other proteolysis, the functional category is unclear. Category is not defined, so I assume this means for any conflicts in annotations, no functional class is assigned. What

if a protein has multiple functional annotations that agree and just one that disagrees? Does this reduce the confidence of the function supported by multiple evidence or are all annotations deemed unreliable? Perhaps I have misunderstood but I think this should be made clearer. Furthermore, there are increasing examples of promiscuous or multifunctional proteins in the literature. How is this possibility handled?

3. The authors have scanned the proteins against Pfam which is a sensible choice as it is a very comprehensive protein family resource. However, I was surprised they hadn't also used InterPro scan. This includes additional resources that could have provided complementary insights. For example, HAMAP which contains information on well curated bacterial families. It might be a good idea to do a final scan of the remaining uncharacterised proteins against InterPro.

4. The authors have not moved any proteins to the probable or confident functional class. I'm confused as to how proteins could be assigned to these classes. As far as I can tell from the introduction proteins were assigned to this class in a previous study if there was a weak confidence match to a TIGRFam family supported by genome context or threading.

Are the authors assuming that this would always give safer annotations than any of the 16 methods that they have applied ie if any of the 16 methods used yielded a highly confident score would that not enable them to move a protein to the probable or confident class? It would be helpful if the authors expanded on this a bit.

Since the authors run their programs on all the proteins it would be interesting to see how many methods do give very high scores for the confident/probable class proteins i.e this could perhaps help in gauging their subsequent assignment of uncharacterised proteins to the probable and putative classes?

Response to reviewers

General response to both reviewers

In response to the reviews we have made substantial changes to the manuscript. We summarise these here before responding to each of the individual comments.

Reviewer 2 made us realise that we could be more confident with the functional assignments we were making, as we were only making assignments up to Putative functions, when the original paper only used three different methods to assign functions and our approach uses many more methods. This was coupled with the suggestion from reviewer 1 (and to some extent by reviewer 2) for the need to validate and benchmark our approach.

We have adjusted our approach by firstly replacing the *Unknown, Generic, Putative, Probable and Equivalog* classes as they combine both the specificity and also the confidence of the annotation, which we think was confusing. Therefore, we have introduced specificity classes that indicate ONLY how specific the annotation is. We use four classes; hypothetical, general, specific and highly specific. At the same time, we consider the confidence of these annotations by the number of methods that have predicted the function and the scores associated with the predictions.

Further, we also assign the broad functional category/biological process that the proteins belong to. Overall, we think this provides a clearer annotation of the functions of the proteins.

Reviewers' comments:

Reviewer #1 (Remarks to the Author):

I have read the manuscript by Wass and coworkers expecting a new methodology or insight in how to assign functions to proteins of unknown function. I must confess I am a bit disappointed. The authors use diverse methodologies and assign generic functions like transport to the majority of proteins and only assign a true functionality to a few. Moreover, there is no validation of the predictions at all, no bootstrapping, benchmark etc..., therefore Without some objective validation I cannot recommend publication of this work in Nature communications.

We first refer the reviewer to the beginning of our response that sets out the significant changes made to the manuscript that are relevant to both reviewers.

We feel that the criticism of Reviewer #1 is based on unrealistic expectations with regard to the current technological and methodological possibilities. Our findings show that the combined use of a wide range of approaches substantially increases the confidence and power of protein function prediction. In addition, our analysis

identifies key challenges that need to be addressed to enable better protein function predictions. Hence, we think that this is a significant contribution to the field. This is important, since less than 1% of proteins in UniProt have experimentally determined Gene Ontology annotations. Due to the advancement of sequencing techniques, more genomic data will become available and the relevance of computational protein function prediction annotations will continue to increase for the foreseeable future.

Additionally, our results demonstrate that the combination of the most advanced protein function prediction methods enable an improved biological understanding. In this case, they provided a new fundamental perspective on the (minimum) prerequisites of life.

We do take the point on a lack of benchmarking. To address this, we have applied our approach to the proteins of known function encoded by the minimal genome (those originally in the putative, probable and equivalog classes) and compared our predictions to the annotations that were available. Note that we have ensured that we removed information that originated from the query sequence when making these predictions. The results are presented on page 10 in the manuscript (text copied below):

Benchmarking our approach using proteins of known function

In contrast to Hutchison et al.¹, who used TIGRFam, genome context and threading to functionally characterise the proteins encoded by the minimal genome, we applied a wider range of approaches to infer their functions. Many methods have been developed to predict protein function using properties ranging from protein sequence to interaction data and predicting features ranging from subcellular localisation to Gene Ontology (GO) terms and protein structure.¹¹ Here, we applied the top performing methods from the recent CAFA^{7,8} assessments, which were available as either a webserver or for download **in combination with** other established methods to assign functions to the proteins encoded by the minimal bacterial genome (see methods and Figure 4). Overall functional inferences were made by manually investigating and combining the predictions and their consistency with genes from the same operon.

To test the performance of our approach, we applied it to the proteins of known function belonging to the Hutchison classes *Putative*, *Probable* and *Equivalog*. For 92% (266 of 289) of the proteins, the functions predicted by our approach agreed with the annotation assigned by Hutchison et al (Figure 3B). Our approach has increased

the confidence of these annotations, with an average of 13 methods making predictions that supported the functional annotations, compared to a maximum of three methods used in the previous study (Figure 3C).

For nine proteins there were minimal differences in the annotations, for example MMSYN1_0637 was previously annotated as the gene *rpsI* which encodes the 30S ribosomal protein S9, whereas our predictions suggest it to be *rpsN* which encodes the 30S ribosomal protein S5 (Table S5), which is probably due to them both belonging to the ribosomal protein S5 domain 2-like superfamily. For 12 proteins, our annotations were less specific than the original ones. These proteins were solely in the Hutchison et al. putative class and the existing annotations were highly specific (Table S5), such as for MMSYN1_0787, our annotation of RelA/SpoT family protein, is more general than the original *relA* gene annotation. For a single protein (MMSYN1_0154) our predicted function of leucyl aminopeptidase was more specific than the initial cytosol aminopeptidase family, catalytic domain protein. Further, only for a single protein (MMSYN1_0908) was our predicted function (*yidC*; inner membrane protein translocase component) completely different to the existing annotation (*misC* - polyketide synthase). Overall, this demonstrates that for proteins with known function our approach is able to assign functions that agree with the existing annotations although in some cases, our assignment may be less specific than the existing annotations. We did not assign functions that disagreed with the known function. Further, with many methods now supporting these functions, there is greater confidence in them.

Reviewer #2 (Remarks to the Author):

This paper describes the application of a combined automatic and manual protocol for assigning protein functions. The approach exploits multiple computational methods and is used to functionally annotate a poorly characterised minimal bacterial genome – less than 1% of the proteins in this genome have experimental characterisation. Previous studies had predicted functions for proteins in this genome, to a reasonable level of confidence, for all but 149 or the 373 proteins. Sixteen different computational methods are applied, ranging from those identifying general features e.g. presence of transmembrane helices to those modelling the structure and predicting the function.

Nearly two thirds of these 149 uncharacterised proteins are assigned a putative function by this integrated computational pipeline with a significant portion being transporter functions most likely reflecting the need for nutrient uptake and waste disposal.

It is a thorough and comprehensive piece of work and the tabulated results represent a significant body of appropriate, rigorous analysis. The methodology is well designed with state-of-the-art resources/algorithms being applied to characterise the proteins in different ways. For each analysis, all the proteins in the minimal genome are analysed and the presentation of these results helps in understanding the extent to which the unknown category differs from the rest. It's interesting that such a high proportion of the proteins in the more confidently assigned functional class can be structurally modelled but so few of the unknown functional class, possibly because they are enriched in membrane proteins. This enrichment in membrane proteins certainly ties in with subsequent analyses suggesting these proteins are transporters.

Points to address

1. I found it difficult to understand what determined how an assignment changed category. In this context some of the statements made are very subjective and unclear. For example;
>For 59 proteins the confidence class was increased. 23 proteins >were assigned additional functional annotations, while for the >remaining 36 proteins the confidence in the existing annotation >was enhanced.

It's not clear to me what determines how confidence is 'increased' or just 'enhanced' and for the 23 that had additional functions if these are different functions does that mean confidence is decreased in the previous functions?

In this context, it would be helpful to have more rigorous definitions of what criteria need to be met to assign a protein from unknown to putative etc. On page 21 the statement:

“If a specific function including a substrate and a biological role was determined and confirmed by multiple sources of >evidence, a protein was classified in the putative confidence >class. Similarly, if a general function was predicted and supported by several lines of evidence, a protein was classified in the generic functional class.”

What is the quantitative difference between multiple and several?

In the sentence below – I'm not sure I understand what is meant by functional category being assigned if 'indicated coherently'.

Does that mean within some threshold set by the method?

We first refer the reviewer to the beginning of our response that sets out the significant changes made to the manuscript that are relevant to both reviewers.

Many thanks for this helpful point that has helped us to improve the clarity of our analysis and manuscript. The section referred to has been reworked and the methods now clearly state how we systematically assigned the protein function to the different specificity classes. The Hutchison et al. classification combined confidence and specificity. From their initial annotations, however, we can only state how specific they are, but we do not know the confidence of their assignment. Hence, we cannot exactly quantify the increase in confidence in the annotations. However, we are able to indicate the confidence levels by the number of methods that support a function and average of the individual confidence scores.

To increase the clarity of our approach, we decided to use our own specificity classes (explained at the beginning of our response) and to separate these from the confidence of the annotation. Our four specificity classes include hypothetical, general, specific and highly specific. The confidence of the annotations is evaluated by the number of methods that predict a function and the average method-associated scores. As a result, much of the text has been rewritten to present the approach in a more systematic fashion (as set out in the methods page – 27 and copied below). We have also included a figure (Figure 4) setting out the prediction process for an individual protein.

As set out above we have altered the classification that we use for the proteins and this should now be clearer. We have provided a set of examples to indicate the level of functional information associated with each of the specificity classes in the text (and further examples in supplementary table 8). The following changes have been

made to clarify our criteria for function annotation in a new section of the paper (page 8) – “Prediction classification considering specificity and confidence”

To infer functions for the proteins of unknown function, we introduced a different way to classify our results, which separates function specificity and prediction confidence. This enabled a more nuanced interpretation of the results than the five classes (*Unknown to Equivalog*) use by Hutchison et al., which combined both specificity and confidence. Our specificity classes included ‘hypothetical’, where the function is completely unknown, ‘general’, where we have some basic functional information (e.g. DNA binding or transporter), ‘specific’, where we have identified a specific function (e.g. transcription factor, ABC transporter) and ‘highly specific’, where a high level of detail is known (e.g. ABC transporter with known substrate; further examples are given in Table S8).

We use the number of methods that support a function and the average score associated with this function as indicators of the confidence of the annotation (see methods). The average score for each predicted function was calculated by normalising the scores from the individual methods (e.g. e-value or probability) to the range of 0-100, with 100 indicating a highly confident score (e.g. a highly significant e-value from Pfam or Gene3D; see methods). Further, each protein was assigned to a larger functional category that represents biological process using the 30 different functional categories proposed by Hutchison et al.

For the GO terms, the authors calculate a combined probability value. I may have missed this, but as far as I could see there were no thresholds on this combined probability value, for assigning the GO term.

It would be helpful if the authors could give a clearer indication of how they used these probability values, presumably in their final manual assessment. The supplementary tables do give an excellent summary of the assignments of each method and are a very useful resource for biologists to assess the likely function and to guide experimentation.

For the combined probability values, we only considered values above 0.65. This ensures that we focus on high confidence functions that have been predicted by the majority of the four GO term-based methods and removes lower confidence predictions that are only made by one or two of the GO term-based approaches. The methods section (part copied below) now sets out how the probability scores are

used. They are used in a second step after initial functions have been identified from a set of the other methods i.e. to support the predicted function. This was seen as the best way to do this as many different GO terms are often predicted, particularly for molecular function terms, many of these are related terms and they can make considering the function more complex.

The methods section (page 27) has been updated to include the following:

The results from the following methods were removed from the analysis if their e-value was above 0.001: TIRGFAM, Pfam, eggNOG-mapper, CathDB FunFams and domains. Models predicted by Phyre2 were kept if the probability of the match was above 80% and e-value was below 0.001. Only results from Firestar with a reliability score above 70% and marked as cognate were retained. Ligands predicted by 3DLigandSite were kept if they were included in at least three homologous models. The best BLAST hit from UniProt (maximum e-value of 0.001) was used to identify the closest homologue of the protein and the information accessible in UniProt was taken into account in the annotation. Additionally, all the predictions of Gene Ontology terms were combined together and the probability of particular terms being predicted by any of the methods were calculated using the following formula: $P(\text{GO}) = 1 - (1 - P(\text{GO}_{\text{FFPred}}))^* (1 - P(\text{GO}_{\text{Argot}}))^* (1 - P(\text{GO}_{\text{CombFunc}}))^* (1 - P(\text{GO}_{\text{LocTree}}))$. **Only high probability (> 0.65) Gene Ontology terms were considered for each of the proteins. For the final prediction of protein function, results from all the methods were manually reviewed. The initial proposition of protein function was based on combining the results from TIRGFAM equivalog families, Pfam domains, InterPro families and domains, eggNOG orthologous groups, CATH functional families, the best BLAST hit from UniProt and the Phyre2 model of the structure. In considering the results from these methods, we looked for agreement between methods, particularly with highly confident results. This initial function was then verified using the predicted Gene Ontology terms and information on predicted ligands (Firestar³⁷, 3DLigandSite³⁸) and transmembrane helices (TMHMM). Where information was not available from the first group of methods, the second group of methods were used as a starting point to infer functions. Transporters and lipoproteins were predicted using membrane transporter (TrSSP) and lipoprotein signal sequences (LipoP) respectively. Finally, it was considered if the predicted function was consistent within a group of genes in the same operon. Where methods made predictions that conflicted with the final predicted function, this was**

noted, but it did not affect the confidence as we recorded the number of methods supporting a function and the average score associated with these predictions (see below).

Confidence of predicted functions was considered for each protein by counting the results that support the final function and calculating the average score from these methods. Results used to calculate the average score come from the methods applied in the first step of function prediction (Figure 4), i.e. TIGRFAM, Pfam, InterPro resources (all but ProSitePatterns), EggNog, Blast, CathFunFams, Phyre2, and also the overall GO term-based prediction (which already combined Argot, CombFunc, FFPred and LocTree) resulting in 17 methods in total. Methods that concern a very specific element of a function, such as transmembrane helices or ligands were not included in the average score calculation. For all methods, scores were normalised to the range 0-100. Most of the methods use e-values as a measure of confidence (e.g. TIGRFAM, Pfam), for these methods $-\log_{10}(\text{e-value})$ was used capping the value at 100. Where probabilities were provided these were multiplied by 100. HAMAP and ProSiteProfiles use scores that are not probabilities or e-values and do not appear to have an upper bound. Considering the scores of these methods for the proteins of known function in the minimal genome indicated that scores were typically in the range 0-100 (Figure S7, S8), so scores above 100 were capped at 100.

2. Regarding conflicts in functional assignments, the authors state on page 21 that if one method points to one category eg transport and the other proteolysis, the functional category is unclear. Category is not defined, so I assume this means for any conflicts in annotations, no functional class is assigned. What if a protein has multiple functional annotations that agree and just one that disagrees? Does this reduce the confidence of the function supported by multiple evidence or are all annotations deemed unreliable? Perhaps I have misunderstood but I think this should be made clearer. Furthermore, there are increasing examples of promiscuous or multifunctional proteins in the literature. How is this possibility handled?

The text on page 21 referred to a single protein, where there were only two methods that made predictions and one pointed to transport and the other proteolysis, in this case the protein functional category was retained as Unclear.

As explained above we start by using a set of the methods to get an initial functional assignment. At this stage, we are looking for consensus between the different methods, particularly where the predictions had high levels of confidence (from the individual methods). It was rare at this initial stage to see conflicting functional predictions and if so they would be by one or two methods and therefore noted but disregarded. This first group of methods are largely sequence/domain based approaches so this may explain why we see considerable agreement between the methods.

As a result of this initial step, we identify a most likely function (note the individual methods may have identified a range of functions at different specificity levels but these can all support an overall prediction). At the second stage, we look for predictions from other methods (GO term-based methods, ligand binding predictions and transmembrane predictions) that support the initial function. The GO term-based approaches sometimes predict multiple different functions –we consider the GO term-based methods to agree if they make predictions above a probability of 0.65 that are compatible with the initial function. Other functions made by the GO term-based methods are then disregarded.

We have added figures (Figure 5 and Figure S6) that indicate the methods that have made predictions for each of the proteins. These show a grey square where a method made a prediction that did not support the overall final prediction. We that there are not many grey squares and they are mainly seen for TsSSP (a method that provides a binary prediction as to whether a protein is a transporter) and 3DLigandSite (which predicts ligand binding sites and associated ligands).

Given our focus on proteins of unknown function, we have considered the most likely function for the proteins. We are aware that proteins can be multifunctional. However for the set of proteins with unknown function, we found that it was difficult to identify a single function and thus we felt it was better to focus on the most likely function from the results of the methods.

3. The authors have scanned the proteins against Pfam which is a sensible choice as it is a very comprehensive protein family resource. However, I was surprised they hadn't also used InterPro scan. This includes additional resources that could have provided complementary insights. For example, HAMAP which contains information on well curated bacterial families. It might be a good idea to do a final scan of the remaining uncharacterised proteins against InterPro.

We have now used InterproScan to identify protein domains/families. We found that HAMAP identified many domains for proteins with known function but few for those of unknown function (data in supplementary Table S7).

4. The authors have not moved any proteins to the probable or confident functional class. I'm confused as to how proteins could be assigned to these classes. As far as I can tell from the introduction proteins were assigned to this class in a previous study if there was a weak

confidence match to a TIGRfam family supported by genome context or threading.

Are the authors assuming that this would always give safer annotations than any of the 16 methods that they have applied ie if any of the 16 methods used yielded a highly confident score would that not enable them to move a protein to the probable or confident class? It would be helpful if the authors expanded on this a bit.

Since the authors run their programs on all the proteins it would be interesting to see how many methods do give very high scores for the confident/probable class proteins i.e this could perhaps help in gauging their subsequent assignment of uncharacterised proteins to the probable and putative classes?

Following this comment, we feel that we were initially too cautious in our assignment of functions. As described above, we now use four new specificity classes. Overall, we found that while many of the specificity changes were from hypothetical to general, there were also changes to the specific and highly specific classes. This is illustrated in the manuscript (page 12):

We assigned a function to 133 of the 149 proteins of unknown function. For nearly half of them (66 of 149), new functional information was provided. This included more specific functions (25), assigning a functional category (5) or both of these (26). For the remaining ten proteins, greater functional information was added but the specificity class or functional category remained the same. For example, MMSYN1_0133 was initially annotated as a peptidase of the S8/S53 family, while we proposed a Subtilisin-like 1 serine protease function. While our annotation is more detailed, it is not highly specific and so the protein remained in the Specific class and Proteolysis functional category.

For 51 proteins, a more specific function was assigned (Figure 5A; Table S5). For 33 proteins that had initially been annotated as hypothetical, a function was now assigned. Twenty-five of these annotations were classified as general, seven as specific and one as highly specific; Figure 5A, Table S5). Eight proteins moved from a general to a specific function (7 specific, 1 highly specific), and 10 proteins were assigned highly specific functions having previously been assigned a specific function (Figure 5A). These predictions vary in their level of confidence. Some of them are supported by many methods, while some have highly confident predictions from a smaller number of methods (Figure 5B,C).

For most proteins that were assigned a general function, we see that they were often supported by fewer methods but those methods predicted them with high confidence scores (Figure 5A). For example, the group of proteins in the bottom right corner of Figure 5C were all predicted to be transporters but only assigned a general function as further details such as substrate specificity could not be inferred. Where specific and highly specific functions were assigned, typically more methods supported the function but there was a greater range in the scores associated from the individual methods (Figure 5). For example, MMSYN1_0298 and MMSYN1_0302 were both initially classed as hypothetical and we have assigned them specific and highly specific functions respectively, based on data available from 10 (MMSYN1_0298) and 12 (MMSYN1_0302) methods (Figure 5; Tables S1-S7). Based on these data sources we propose that MMSYN1_0298 is a ribosomal protein from the family L7AE/L30e (Figure 6A) and that MMSYN1_0302 is an oxygen-insensitive NAD(P)H nitroreductase (Figure 6B), both of which are functions widespread across the kingdoms of life.

And also on page 15:

For the remaining 83 proteins, our predictions supported the existing annotation. Importantly for many of these proteins, multiple methods have now made predictions that support the annotation, thus increasing their confidence. Figure 7 shows that many of the proteins (28 out of 83) have predicted functions that are supported by 10 or more methods, rising to 61 supported by 5 or more methods, often with high confidence scores (or e-values) from the individual methods.

Our approach in assigning function was to look for a consensus between methods. Our results show that for the proteins where we increase their specificity class that there is a range in the number of methods that make predictions supporting those functions and the confidence of the results from those individual methods (Figure 5C). We typically found for these proteins that where only a few methods made predictions, they were for general functions (i.e. group of points at bottom of right of figure 5C) and that where we were able to infer more specific functions, then more methods made supporting predictions and the individual confidence scores varied from 25-100 (our normalised scores – Figure 5B and 5C).

To benchmark our approach, we assigned functions to all of the proteins of known function. We saw a strong level of agreement with the existing functions annotated

by Hutchison et al., Our predictions are typically made by many methods (average of 14 – figure 3C). We have added the following the text to the manuscript (page 10):

Benchmarking our approach using proteins of known function

In contrast to Hutchison et al.¹, who used TIGRFam, genome context and threading to functionally characterise the proteins encoded by the minimal genome, we applied a wider range of approaches to infer their functions. Many methods have been developed to predict protein function using properties ranging from protein sequence to interaction data and predicting features ranging from subcellular localisation to Gene Ontology (GO) terms and protein structure.¹¹ Here, we applied the top performing methods from the recent CAFA^{7,8} assessments, which were available as either a webserver or for download **in combination with** other established methods to assign functions to the proteins encoded by the minimal bacterial genome (see methods and Figure 4). Overall functional inferences were made by manually investigating and combining the predictions and their consistency with genes from the same operon.

To test the performance of our approach, we applied it to the proteins of known function belonging to the Hutchison classes *Putative*, *Probable* and *Equivalent*. For 92% (266 of 289) of the proteins, the functions predicted by our approach agreed with the annotation assigned by Hutchison et al (Figure 3B). Our approach has increased the confidence of these annotations, with an average of 13 methods making predictions that supported the functional annotations, compared to a maximum of three methods used in the previous study (Figure 3C).

For nine proteins there were minimal differences in the annotations, for example MMSYN1_0637 was previously annotated as the gene *rpsI* which encodes the 30S ribosomal protein S9, whereas our predictions suggest it to be *rpsN* which encodes the 30S ribosomal protein S5 (Table S5), which is probably due to them both belonging to the ribosomal protein S5 domain 2-like superfamily. For 12 proteins, our annotations were less specific than the original ones. These proteins were solely in the Hutchison et al. putative class and the existing annotations were highly specific (Table S5), such as for MMSYN1_0787, our annotation of RelA/SpoT family protein, is more general

that than the original relA gene annotation. For a single protein (MMSYN1_0154) our predicted function of leucyl aminopeptidase was more specific than the initial cytosol aminopeptidase family, catalytic domain protein. Further, only for a single protein (MMSYN1_0908) was our predicted function (yidC; inner membrane protein translocase component) completely different to the existing annotation (misC - polyketide synthase). Overall, this demonstrates that for proteins with known function our approach is able to assign functions that agree with the existing annotations although in some cases, our assignment may be less specific than the existing annotations. We did not assign functions that disagreed with the known function. Further, with many methods now supporting these functions, there is greater confidence in them.

Reviewers' Comments:

Reviewer #1:

Remarks to the Author:

My main concern remains on the validity of the approach. The hypothesis behind the authors method is that by using many different tools and combining the normalized prediction they could get more accurate assignment. A problem is how many of these methods have a similar underlying methodology and therefore are just duplicates of the same prediction method with different nuances. If they do their global analysis and now they look at individual methods is there any of them that agrees well with the global one?,

I think the authors have been thorough and rigorous in their analysis, but at the end of the day have they found some unexpected new functionality that could explain some exciting biology of the minimal genome?, Aside from some small differences do they offer anything fundamentally new from the assignment done by Danchin ?

Reviewer #2:

Remarks to the Author:

The authors have addressed all my concerns and performed additional analyses that have considerably strengthened the manuscript.

Reviewer #1 (Remarks to the Author):

My main concern remains on the validity of the approach. The hypothesis behind the authors method is that by using many different tools and combining the normalized prediction they could get more accurate assignment. A problem is how many of these methods have a similar underlying methodology and therefore are just duplicates of the same prediction method with different nuances. If they do their global analysis and now they look at individual methods is there any of them that agrees well with the global one?,

We have compared the predictions made by the individual methods to the overall predictions made through the combination of all the methods. We find that at most one-third of the predictions of any individual method support the overall prediction to the same level of detail. Further, if we take the top five of the methods, they all work in different ways and any combination of pairs of these five obtains the final annotations assigned for only 25% of the proteins. Together, our analysis demonstrates the improvement obtained by combining multiple methods together.

We have added the following to the text to the results section (from line308) to address this and two supplementary tables (Table S7, S8):

Overall, we found that the diversity of different methods used was required for inferring function, with no individual method able to predict the most detailed function assigned to more than one-third of the proteins of unknown function (Table S7). The top five methods to assign the most detailed functions each used different approaches, including a method that identifies orthologous groups (EggNOG¹²), the group of methods that predict GO terms, a method that predicts protein three-dimensional structure (Phyre2¹⁰), identification of protein domains from Pfam and finally the best BLAST match from UniProt. Further, any combination of the top five performing methods only obtained the final annotation for a maximum of 25% of the proteins, further highlighting the contribution of multiple different methods to assign functions (Table S8). Two methods (GO terms and TMHMM) were able to widely provide more generic functions supporting the overall assigned function (54% for GO terms and 82% for TMHMM), although TMHMM only predicts if the protein contains transmembrane helices (Table S7).

The new supplementary tables (S7, S8) are shown on the next page.

Method	Yes (final)	Yes (general)	No	No prediction	Yes (final)	Yes (general)	No	No prediction
EggNog	55	22	1	71	37%	15%	1%	48%
GO Terms	53	80	0	16	36%	54%	0%	11%
Phyre2	53	32	0	64	36%	21%	0%	43%
Blast against UniProt top match	51	20	1	77	34%	13%	1%	52%
Pfam	49	34	0	66	33%	23%	0%	44%
CathDB FunFams	45	16	2	86	30%	11%	1%	58%
TIGRFAM	41	24	1	83	28%	16%	1%	56%
InterPro ProSiteProfiles	21	12	0	116	14%	8%	0%	78%
InterPro CDD	21	21	0	107	14%	14%	0%	72%
InterPro SUPERFAMILY	21	40	0	88	14%	27%	0%	59%
TrSSP	14	71	48	16	9%	48%	32%	11%
InterPro Gene3D	14	28	1	106	9%	19%	1%	71%
InterPro PIRSF	7	4	0	138	5%	3%	0%	93%
InterPro Hamap	7	1	0	141	5%	1%	0%	95%
InterPro SMART	7	11	0	131	5%	7%	0%	88%
TMHMM	6	122	5	16	4%	82%	3%	11%
InterPro ProSitePatterns	4	12	0	133	3%	8%	0%	89%
InterPro PRINTS	3	4	0	142	2%	3%	0%	95%
InterPro SFLD	2	1	0	146	1%	1%	0%	98%
3DLigandSite	0	44	20	85	0%	30%	13%	57%
Firestar	0	35	2	112	0%	23%	1%	75%
InterPro ProDom	0	1	0	148	0%	1%	0%	99%

Table S7. Comparison of the predictions made by individual methods and the final annotation assigned by the combination of methods. For each individual method we counted the predictions that agreed with the final annotation assigned to the protein (column yes – final) and if they more generally agreed with the assigned function (yes – general).

Method 1	Method 2	Number of common proteins	Percentage
EggNog	Blast - UniProt	38	25.5
EggNog	Pfam	31	20.81
EggNog	Phyre2	30	20.13
Phyre2	Pfam	28	18.79
Phyre2	Blast - UniProt	26	17.45
Blast - UniProt	Pfam	24	16.11
EggNog	GO Terms	14	9.4
GO Terms	Phyre2	13	8.72
GO Terms	Blast - UniProt	12	8.05
GO Terms	Pfam	9	6.04

Table S8. Common predictions made by the five methods with greatest agreement with the final annotation. For each pair of methods the number of proteins where both methods make the same prediction as the final annotation is shown.

We have also added the following sentence to the discussion (from line 474):

Our analysis shows that the combination of many methods was essential with no single method able to identify the highest detailed function assigned to more than one-third of the proteins (Table S7).

I think the authors have been thorough and rigorous in their analysis, but at the end of the day have they found some unexpected new functionality that could explain some exciting biology of the minimal genome? Aside from some small differences do they offer anything fundamentally new from the assignment done by Danchin ?

The proportion of the proteins of unknown function that we make functional assignments to is much greater than that done by the Danchin and Fang paper. We further highlight 1) that our approach is completely different to Danchin/Fang and 2) that little detail of the methods used by Danchin are available (there is no methods section in the Danchin and Fang paper) making it difficult for others to reproduce. Our approach could readily be adopted by others.

To demonstrate the difference in our assignments compared to those by Danchin and Fang we have added the following paragraph to our comparison (from line 431- new text in red):

Comparison of the results from both studies revealed considerable overlaps (Table S9). Using our approach, only sixteen proteins remained hypothetical without any assigned function, while Danchin and Fang did not provide any annotations for 78 of the proteins with unknown function. Thus, we leave only 10% of the previously unannotated proteins without any assigned function, while 52% remain completely uncharacterised by Danchin and Fang. This demonstrates the breadth of function that our approach is able to assign.

Reviewer #2 (Remarks to the Author):

The authors have addressed all my concerns and performed additional analyses that have considerably strengthened the manuscript.

We thank the reviewer for the insightful comments that assisted us in improving the manuscript.